# Characterization of Antagonistic Bacteria *Paenibacillus polymyxa* ZYPP18 and the Effects on Plant Growth

**DOI:** 10.3390/plants12132504

**Published:** 2023-06-30

**Authors:** Xiangying Li, Sujing Ma, Yuan Meng, Wei Wei, Chen Peng, Chunli Ling, Susu Fan, Zhenyu Liu

**Affiliations:** 1College of Plant Protection, Shandong Agricultural University, Taian 271018, China; nongdaxiangyingli@126.com (X.L.); masunshine0811@163.com (S.M.); mengyuan20210226@163.com (Y.M.); 18354819073@163.com (W.W.); 18853886816@163.com (C.P.); 2Ecology Institute, Qilu University of Technology (Shandong Academy of Sciences), Jinan 250014, China; lingchunli1999@163.com

**Keywords:** *Paenibacillus polymyxa*, plant growth promotion, antagonism, wheat sheath blight

## Abstract

*Paenibacillus polymyxa* is a plant growth–promoting rhizobacteria (PGPR) that has significant biocontrol properties. Wheat sheath blight caused by *Rhizoctonia cerealis* is a significant soil–borne disease of wheat that causes significant losses in wheat production, and the biological control against the disease has received extensive attention. *P. polymyxa* ZYPP18 was identified using morphological and molecular characterization. An antagonistic activity experiment verified that ZYPP18 inhibits the growth of *R. cerealis* on artificial growth media. A detached leaf assay verified that ZYPP18 inhibits the expansion of wheat sheath blight on the detached leaf. ZYPP18 has been found to possess plant growth–promoting properties, as well as the ability to solubilize phosphate and generate indole–3–acetic acid. Results from hydroponic experiments showed that wheat seedlings treated with ZYPP18 grew faster. Additionally, pot experiments and field experiments demonstrated that ZYPP18 effectively controls the occurrence of wheat sheath blight. ZYPP18 reduced the incidence of wheat sheath blight in wheat seedlings by 37.37% and 37.90%, respectively. The control effect of ZYPP18 on wheat sheath blight was 56.30% and 65.57%, respectively. These findings provide evidence that *P. polymyxa* ZYPP18 is an effective biological factor that can control disease and promote plant growth.

## 1. Introduction

Plant growth–promoting rhizobacteria (PGPRs) have received increasing attention for their plant disease control ability and environmentally friendly potential [1,2]. PGPRs were verified to benefit plant root growth and produce antimicrobial substances, making them essential in the context of biocontrol [3]. PGPRs enhance plant growth through multiple direct or indirect mechanisms, including gibberellin, auxin, and indole acetic acid auxin synthesis, biological nitrogen fixation and phosphate dissolution [4,5], organic acid production and reduction of Ethylene content, and increased root density and length through 1–aminocyclopropane 1–carboxylate deaminase (ACC) production [6,7]. Numerous plant growth–promoting rhizobacteria (PGPR) are known to biosynthesize and exude defense–related metabolites that serve to trigger systemic resistance (ISR) within the host plant. In addition, PGPRs can suppress pathogen growth through multiple modes of action, including direct competition for nutrients and niches, enzyme lysis, antibiosis, and signal interference [8,9,10].

Wheat is one of the three major crops and is the main food staple [11]. Wheat sheath blight caused by *Rhizoctonia cerealis* is a significant soil–borne disease of wheat, which infects wheat roots and causes stems to shrink [12]. The predominant method of controlling soil–borne diseases, such as wheat sheath blight, typically involves the application of chemical fungicides, which may result in environmental pollution and the emergence of fungal resistance [13]. Biological control involves using antagonistic microbes and metabolites to control the disease, demonstrating excellent effects and impressive prospects [14].

The genus *Paenibacillus* is Gram–positive and sporulating. With over 150 species identified, this genus holds paramount importance in agriculture as PGPRs [15,16]. *Paenibacillus polymyxa* is one of the most critical species out of all of these species [17].

*P. polymyxa* has been reported to enhance plant growth and disease resistance through the biosynthesis and secretion of plant growth hormones, such as dissolved inorganic phosphates and indole–3–acetic acid (IAA) [15,18,19,20,21,22,23]. Many of the strains of *P. polymyxa* can directly inhibit the plant pathogen, thereby protecting the plant from pathogen infection, mainly due to the production of antibiotic compounds such as polymyxins, hydrolytic proteases with glycosyl groups, fusaricidin polypeptide antibiotics, and beta–glucans [24,25,26,27]. The valuable properties exhibited by *P. polymyxa* strains have garnered significant interest in their potential application for biofertilization and biocontrol development [28].

In this study, a novel strain of *P. polymyxa* was isolated and subjected to biocontrol assays. Results indicated that the strain exhibited significant plant growth–promoting effects on wheat and demonstrated potent biocontrol activity against wheat sheath blight disease. Collectively, these findings suggest that the identified *P. polymyxa* strain holds significant promise as a viable biocontrol agent in agricultural practices.

## 2. Results

### 2.1. Bacterium Identification

One isolate with a distinct inhibitory effect on *R. cerealis* was picked out for further study. The isolate formed light pale yellowish colonies with a thick central part surrounded by a glistening visible part on the LA plate (Figure 1A). The isolate was a Gram–positive and spore–forming bacterium and exhibited small rod–shaped structures typical of the genus *Paenibacillus* (Figure 1B,C). The isolate showed alignment as ZYPP18. The 16S rRNA, *gyrA*, *and rpoB* genes were cloned and sequenced. The partial sequences of the three genes were compared to the sequences using the BLAST search program in NCBI’s GenBank, which showed high correlations with the genes of the species belonging to *Paenibacillus* spp. Three phylogenetic trees were constructed using MEGA 11 with the partial sequences of 16S rRNA, *gyrA*, and *rpoB* of ZYPP18 and other typical strains of *Paenibacillus* spp. (Figure 1D–F).

### 2.2. Antagonistic Activity

#### 2.2.1. Inhibition Effect on *R. cerealis*

*P. polymyxa* ZYPP18 had strong antagonistic activity against *R. cerealis,* with an inhibition rate of 92.68% (Figure 2). The fermentation filtrate of ZYPP18 also inhibited the mycelial growth of *R. cerealis* (Figure 3). When treated with the fermentation filtrate of ZYPP18 cultured for 3 d, 4 d, and 5 d, at 1/2 concentration, the inhibition rates were 45.73%, 46.45%, and 48.90%, respectively (Figure 3A,B).

#### 2.2.2. Disease Inhibition Effect on Detached Leaves

The experiment was conducted by inoculating *R. cerealis* on detached wheat seedling leaves sprayed with either sterile water or *P. polymyxa* ZYPP18. The control group treated with sterile water showed a complete susceptibility to *R. cerealis*, as evidenced by lesion expansion with an average length of 2.50 cm after 5 days of inoculation. In contrast, the group treated with *P. polymyxa* ZYPP18 exhibited mild to no infection by *R. cerealis*, with an average lesion length of only 0.075 cm after the same duration. This clearly indicates the remarkable inhibitory effect of *P. polymyxa* ZYPP18 against wheat sheath blight disease, as depicted in Figure 4.

#### 2.2.3. Detection of Genes Related to Antibacterial Substance Synthesis

The partial fragmentation genes encoding the antibiotics polymyxin C (*PMXC*), polymyxin D (*PMXD*), Fusaricidins (*fusA*), β–glucanase (*PJT*), hydrolysis of proteases CEL44C–MAN26A (*cel44C*), and Cellulase A (*cel5A*) and Cellulase B (*cel5B*) were detected by PCR amplification (Figure 5). It indicates that *B. polymyxa* ZYPP18 possesses the genes related to antibacterial substance synthesis, implying the function of antibacterial substances in the biological effect.

### 2.3. Plant Growth Promotion

#### 2.3.1. Growth–Promoting Effect on Wheat Seedlings

Wheat treated with a fermentation filtrate and bacteria of *P. polymyxa* ZYPP18 exhibited significant growth–promoting effects (Figure 6). ZYPP18 had discernible impacts on multiple phenotypic traits related to wheat seedling growth, including plant height, stem diameter, leaf length, and leaf width (Figure 6B–E). Our findings indicate that ZYPP18 did not elicit noticeable changes in wheat plant height. However, treatment with 4 mL of fermentation filtrate led to a significant increase in wheat leaf length, while no significant effects were observed in leaf width and the remaining parameters. Notably, treatments with 100 μL, 1 mL, 2 mL of fermentation filtrate, and 2 μL of bacterial suspension successfully promoted wheat stem diameter growth. The fermentation filtrate treatment and bacteria of ZYPP18 benefited the growth of the wheat root (Figure 6F–I). All the treatments with different volumes of fermentation filtrate and bacteria of ZYPP18 significantly promoted the growth of root length, root surface area, and root volume. As the volume of fermentation filtrate was increased, it was observed that the promoting effect on wheat roots also increased. Notably, a relatively weaker promoting effect was observed upon treatment with 4 mL of fermentation filtrate. It is speculated that the deleterious impact of fermentation filtrate on root growth may become more pronounced at higher volumes exceeding 4 mL. The findings of our study underscore the capacity of ZYPP18 to promote the growth of wheat, with a particular emphasis on root development.

#### 2.3.2. Phosphorus Solubilization Activity

As shown in Table 1, *B. polymyxa* ZYPP18 can dissolve phosphate, which could dissolve organic phosphorus or inorganic phosphorus. The ratio of the transparent circle to colony diameter of ZYPP18 (D/d) was 1.4 in the organic phosphorus medium and 1.6 in the inorganic medium (Table 1).

The enzyme activity results provide evidence of the phosphate–releasing ability of *B. polymyxa* ZYPP18 (Table 2). Phosphatase activity, including acid phosphatase, alkaline phosphatase, and neutral phosphatase, was detected in the cultures of ZYPP18 cultured for 3 d and 5 d.

#### 2.3.3. IAA Production

For L–Ser–inducing IAA production, the IAA contents tested were 3.66 mg/mL in the R2A medium with 200 mg/L L–tryptophan, 3.05 mg/mL in the LB medium with 200 mg/L L–tryptophan, and 2.74 mg/mL in LB (Figure 7A). The gene that encodes IAA in ZYPP18 was detected by PCR amplification, and a band of approximately 1800 bp was detected (Figure 7B). Therefore, *B. polymyxa* ZYPP18 possesses the IAA synthesis gene and can produce IAA.

### 2.4. Control Effect of ZYPP18 on Wheat Sheath Blight in Pot and Field Experiments

The pot experiment showed that the disease incidence of wheat seedlings treated with *R. cerealis* was 36.90%, while it was 23.00% when treated with *R. cerealis* and ZYPP18 (Figure 8A). Inoculation with ZYPP18 reduced the incidence of wheat sheath blight in wheat seedlings by 37.37%. The disease index of wheat seedlings treated only with *R. cerealis* was determined to be 40.11, whereas the application of ZYPP18 in conjunction with *R. cerealis* significantly decreased the disease index to 17.53 (Figure 8B). The observed control efficacy of ZYPP18 against wheat sheath blight was estimated to be 56.30% (Table 3).

The field experiment showed that the disease incidence of wheat seedlings treated with *R. cerealis* was 61.69%, while it was 38.31% when treated with *R. cerealis* and ZYPP18 (Figure 8C). Inoculation with ZYPP18 reduced the incidence of wheat sheath blight in wheat seedlings by 37.90%. The disease index of wheat seedlings treated with *R. cerealis* was 59.75, while it was 20.57 when treated with *R. cerealis* and ZYPP18 (Figure 8D). The control effect of ZYPP18 on wheat sheath blight was 65.57% (Table 3).

## 3. Discussion

Biological control represents an efficacious method for managing plant diseases, which leverages the capabilities of beneficial microorganisms and biological metabolites to suppress plant pathogens, improve plant immunity, and modify the plant growth environment [29,30,31]. In this study, *P. polymyxa* ZYPP18 was verified as being such a microorganism, possessing the dual capacity of producing antibiotic substances and promoting plant growth.

*P. polymyxa* is widely isolated from the rhizosphere soils of many plants, including crops such as wheat [32] and rice [33]; vegetables such as tomatoes [34], beans [35], peppers, and cucumber beans [36]; and other plants such as maize [37], sunflower [38], and Dendrobium [39]. *B. polymyxa* has been demonstrated to possess the ability to suppress numerous plant diseases [40]. For example, *P. polymyxa* HX–140 inhibits the growth of the pathogen of cucumber wilt disease [41]. In addition, *P. polymyxa* Nl4 effectively controlled pear *Valsa canker* caused by *Valsa pyri* [42], while *P. polymyxa* Y–1 controlled rice bacterial disease [43]. To date, there have been no studies reporting on the utilization of *P. polymyxa* strains for the purpose of biological control of wheat sheath blight. In this study, ZYPP18, a biocontrol bacterium isolated and screened from the rhizosphere soil of healthy tobacco, had an obvious inhibitory effect on wheat sheath wilt through plant tests, including an in vitro leaf experiment, a pot experiment, and a field experiment. The field control effect of ZYPP18 on wheat sheath blight was 65.57%. Therefore, it can be exploited as an environmentally biological agent or microbial–friendlier alternative to chemical fertilizers.

*P. polymyxa* is the main microbial population for the biological control of plant diseases, owing to its notable features such as high stress resistance, production of a variety of lipopeptide antibiotics, and broad bacteriostatic spectrum [23,24,25,27]. We revealed that *P. polymyxa* ZYPP18 had strong antagonistic activity against *R. cerealis* with an inhibition rate of 92.68%. The fermentation filtrate of ZYPP18 also inhibited the mycelial growth of *R. cerealis*. Several genes related to antibacterial substance synthesis were detected in ZYPP18, including the genes encoding the antibiotics Polymyxin, Fusaricidins, β–glucanase, hydrolysis of proteases CEL44C–MAN26A, and cellulase. This indicates that *P. polymyxa* ZYPP18 has multiple disease–resistant mechanisms.

IAA plays a crucial role in the establishment of a stable seedling root system by promoting the development of secondary roots. Conversely, bacteria with the function of solubilizing phosphorus secrete organic acids and some metal ions to accrete and convert the insoluble phosphorus in the soil into soluble phosphorus, which can be absorbed by plants and promotes plant growth. Notably, genome analysis has revealed that the majority of *P. Polymyxa* strains harbor genes involved in gluconic acid synthesis and that encode glucose–1–dehydrogenase and gluconate dehydrogenase, which aid in the production of gluconate–dissolved phosphorus [28]. Weselowski [28] reported *that P. polymyxa* CR1 could dissolve inorganic phosphates. Moreover, *P. polymyxa* harbors a gene responsible for the synthesis of IAA precursors and which encodes aminotrans–ferase L–tryptophan for oxidative deamination to indole–3–pyruvate (IPA), a crucial intermediate for IAA biosynthesis [44,45]. In this regard, ZYPP18 exhibited proficient IAA production and decarboxylation of phosphorus. Notably, the promoting effect of the ZYPP18 fermentation filtrate on wheat growth was positively correlated with the concentration of filtrate. However, excess filtrate concentrations compromised wheat growth, possibly due to high IAA concentration–induced growth inhibition. Thus, an optimal filtrate concentration is pivotal for obtaining the maximum promotive potential of ZYPP18 on wheat growth.

## 4. Materials and Methods

### 4.1. Isolation and Identification of the Strain

Soil samples were collected from a tobacco field in Zhucheng City, Shandong Province, China. Dilution plate techniques were employed to isolate the bacteria. Soil samples were serially diluted with ten–fold dilutions up to 10^−5^ in sterile water. A total of 100 μL of soil diluent were plated on LA (10 g tryptone, 5 g yeast powder, 10 g NaCl, 15 g agar, and 1000 mL deionized water) plates and cultured at 28 °C. The preferred isolates were picked out and cultured on LA plates to observe their morphological characteristics.

The bacteria cultured in LB were collected, and genomic DNA was extracted using an OMEGA bacterial DNA kit for amplified 16S rRNA. The 16S rRNA, *gyrA* gene, and *rpoB* gene fragments were PCR amplified using universal bacterial primers. The primers for the 16S rRNA gene were 27f (5′–AGAGTTTGATCCTGGCTCAG–3′) and 1492r (5′–CGGTGTGTACAAGGCCC–3′) [46]; the primer pair for gyrA1 was *gyrA*–F (5′–CAGTCAGGAAATGCGTACGTCCTT–3′) and *gyrA*–R (5′–CAAGGTAATGCTCCAGGCATTGCT–3′); and the primer pair for r*poB* was *rpoB*_2292f–F (5′–GACGTGGGATGGCTACAACT–3′) and *rpoB*_3354r (5′–ATTGTCGCCTTTAACGATGG–3′) [47]. The 16S rRNA, *gyrA* gene, and *rpoB* gene sequences obtained were compared to the sequences using the BLAST search program in NCBI’s GenBank (https://www.ncbi.nlm.nih.gov/) (accessed on 5 January 2022). The phylogenetic tree was constructed using MEGA 11, and Maximum Likelihood was used [48,49].

### 4.2. In Vitro Antagonism Test against R. cerealis

#### 4.2.1. Antimicrobial Activity of Antagonistic Bacteria

The strain of *R. cerealis* was isolated from the samples of wheat sheath blight and kept in the Key Laboratory of Agricultural Microbiology, Shandong Agricultural University. A 5 mm dish of *R. cerealis* was placed onto the center of a PDA plate, and then 2 μL of bacterial suspension (≈1 × 10^8^ CFU/mL) were dropped onto the same petri dish 2 cm away from the dish of *R. cerealis,* with the one without drops of bacterial suspension used as a control. Treatments and controls were replicated three times. The plates were incubated at 28 °C. When the colony of the control was just full–grown on the PDA plate, then the fungal colony radius was measured. The inhibition rate was calculated as follows. Inhibition rate (%) = (radius of control fungal colony − radius of treated fungal colony)/radius of control fungal colony × 100%.

#### 4.2.2. Antimicrobial Activity of Fermentation Filtrate

The antagonistic bacteria were cultured in 150 mL LB media at 28 °C and 150 rpm for 3, 4, and 5 days. The fermentation filtrate was obtained by filtration with a 0.22 μm sterile membrane. The PDA (200 g potato, 15 g glucose, 18 g agar, and 1000 mL water) and 2 × PDA (200 g potato, 15 g glucose, 18 g agar, and 500 mL water) were prepared. The fermentation filtrate was added to the 2 × PDA media with a volume ratio of 1:1, 1:3, 1:5, and 1:9, and replenished to the same volume with sterile water. A 5 mm dish of *R. cerealis* was placed onto the center of a PDA plate with different concentrations of fermentation filtrate and cultured at 28 °C. The fungal dish incubated onto the PDA medium without the fermentation filtrate was treated as a control. Treatments and controls were replicated three times. The diameter of the fungal colony was measured when the colony of the control was just full–grown on the PDA plate, and the inhibition rate was calculated using the following formula. Inhibition rate (%) = (diameter of control fungal colony − diameter of treated fungal colony)/diameter of control fungal colony × 100%.

#### 4.2.3. Detached Leaves Experiment

The detached leaves of wheat seedlings were used to determine the effect of the bacteria on wheat sheath blight by measuring the disease expansion on the leaf surface. A 5 cm length of leaf from the second wheat leaf of the seedling was cut and sterilized with alcohol. The bacteria were grown in LB at 28 °C at 180 r/min for 24 h, and the bacterial suspension was collected. The pellet was resuspended in sterile water and adjusted to 1 × 10^8^ CFU/mL. The bacterial suspension was sprayed on the detached leaf, then a 5 mm fungal dish of *R. cerealis* was inoculated on the center of the leaf surface. Next, the leaves were incubated in Petri dishes at 25 °C, and the disease lesion was measured daily. The detached leaves sprayed with sterile water were treated as a control. Treatments and controls were replicated four times.

#### 4.2.4. Detection of the Genes Related to Antimicrobial Substance Synthesis

The bacteria cultured in LB were collected, and genomic DNA was extracted using an OMEGA bacterial DNA extraction kit according to the manufacturer’s instructions. The primers for detection of the genes related to antibacterial substance production, including Polymyxin C, Polymyxin D, Fusaricidins, and hydrolysis of proteases, β–glucanase, and cellulases, were designed using Primer 5.0 [50]. All the primers are listed in Table 4. PCR was performed using the thermal cycle to detect the genes associated with antimicrobial substance synthesis.

### 4.3. Growth–Promoting Properties Test

#### 4.3.1. Plant Growth–Promoting Experiments

A plant growth–promoting experiment was performed with a hydroponic experiment. The wheat seeds were sterilized for 10 min with 1% NaClO, washed three times with sterile water, and incubated at 25 °C for germination. When the seeds germinated and grew to 1 cm high, the seedlings were transplanted into pots with 500 mL Hoagland nutrient solution (945 mL Ca(NO_3_)_2_·4H_2_0, 607 mL KNO_3_, 115 mL NH_4_HPO_4_, 493 mL MgSO_4_·7H_2_O, 2.5 mL Fe–citrate, and 5 mL micro–elements). Next, 10 μL, 100 μL, 1 mL, 2 mL, and 4 mL of fermentation filtrate and 2 μL of bacteria were inoculated in the nutrient solution, respectively. Ten wheat seedlings were planted in each pot, and three pots were treated as treatments and incubated in a light incubator with a 14 h light/10 h dark cycle at 25 °C. The seedlings’ height, total root length, leaf length, and leaf width were measured using a ruler. The total root length, root volume, and root surface area were measured using a root scanner (WinRHIZO, AgriPheno, Shanghai, China).

#### 4.3.2. Phosphate Solubilization Assay

For detection of the phosphorus solubilization activity, the bacteria were cultured on the Pikovskaya (PVK) inorganic phosphorus medium and the Mongina medium for 7 d at 28 °C.

The PVK and Mongina mediums were prepared as follows: PVK medium, 10 g glucose, 0.5 g (NH_4_)_2_SO_4_, 0.2 g NaCl, 0.2 g KCl, 0.03 g MgSO_4_·7H_2_O, 0.03 g MnSO_4_, 0.003 g FeSO_4_, 0.5 g yeast extract, 10 g agar, 1000 mL distilled water, and pH = 7.0; Ca_3_(PO_4_)_2_ was sieved, sterilized separately, and then mixed with the medium; Mongina medium, 10 g glucose, 0.5 g (NH_4_)_2_SO_4_, 0.3 g MgSO_4_·7H_2_O, 0.3 g NaCl, 0.3 g KCl, 0.36 g FeSO_4_·7H_2_O, 0.03 g MnSO_4_·H_2_O, and 5 g CaCO_3_, 18 g agar.

The bacteria were cultured in the Mongina medium for 3 d and 5 d at 28 °C, 150 r/min. Determination of bacterial phosphatase activity was measured spectrophotometrically by the disodium phenyl phosphate method of Li [51].

The bacteria were incubated in the acidic acetic acid–sodium acetate buffer with pH 4.6, pH 7.0, and pH 9 for 24 h; then the OD value was measured at OD_570_. The activity was calculated by calculating the amount of phenol production in 1 mL of bacterial solution after 24 h. All assays were carried out in triplicate.

#### 4.3.3. IAA Production Assay

The bacteria were cultured in R2A and LB with 200 mg/L L–tryptophan at 28 °C for 3 days, and then IAA quantification was carried out using the Salkowski method [52]. The bacteria cultured in LB IAA were quantified as well. The non–inoculated mediums were conducted at 28 °C for 3 days and used as negative controls. The bacteria were centrifuged at 10,000 r/min, and then the supernatant was mixed with the Salkowski reagent. The optical density was measured at 530 nm. The experiment was repeated thrice. A standard curve was generated with the optical densities for 0, 5, 10, 20, 40, and 60 mg/L of indole acetic acid.

#### 4.3.4. IAA Gene Detection

The bacteria were cultured in LB, and the genomic DNA was extracted as mentioned above. The gene encoding IAA was detected by PCR amplification using primers of primer1 (GGGAATTCTTACTCGTCCCCCATCAGC) and primer2 (CTCGGATCCCCAATGAGTGCACAAATTCC) [21].

### 4.4. In Vivo Plant Experiment

#### 4.4.1. Pot Experiment

Sterilized wheat seeds were inoculated with 5 fungal disks of *R. cerealis* and cultured at 28 °C for 20 d to obtain the inoculant of the wheat seeds. The wheat seeds were sterilized for 10 min with 1% NaClO, washed three times with sterile water, and incubated at 25 °C for germination. When the seeds germinated and grew to 1 cm high, the seedlings were transplanted into the pots. Twenty wheat seedlings were planted in each pot, and three pots were treated as treatments and incubated at 25 °C; 20 days after, 100 mL sterile water with 500 μL bacteria (about 1 × 10^8^ CFU/mL) were inoculated into the soil in the pot. The wheat seed inoculants were inoculated near the wheat roots. No–bacterial treatment and inoculating with inactivated wheat seed inoculant served as controls. Twenty days after, the disease incidence and disease index were investigated.

The disease incidence of wheat was investigated according to the method of Youssef [53]. Disease incidence = (number of diseased plants/total number of investigated plants) × 100%; Disease index = Σ (number of diseased plants at all levels × representative of all levels)/(total number of plants under investigation × the highest level of representative value) × 100; Control effect = (control disease index − treatment of disease index)/control disease index × 100%.

#### 4.4.2. Field Experiment

The field plot employed in the experiment was 1.5 m wide and 5 m long. Wheat seeds were sowed in soil with a 4–5 cm depth and six rows were planted. Each treatment was replicated three times. Two days after sowing, 1mL bacteria were inoculated into the soil (about 1 × 10^8^ CFU/mL) per row. Twenty days after, 10 g of wheat seed inoculant were inoculated in each row. No–bacterial treatment and inoculating with inactivated wheat seed inoculant served as controls. After 20 d of inoculation with the wheat seed inoculant, the diseases were surveyed as per the methods in the pot experiment.

### 4.5. Statistical Analysis

GraphPad Prism 8 software was used for data collation and graphing. One–way ANOVA (*p* < 0.05) was performed using IBM SPSS Statistics 20 software, and Duncan’s and Dunnett’s T3 methods were used for multiple comparisons. Marking of significant differences.

## Figures and Tables

**Figure 1 plants-12-02504-f001:**
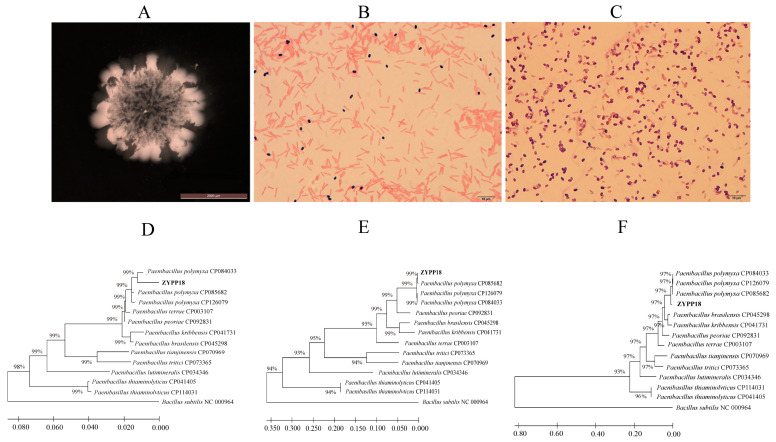
Bacterium identification of *P. polymyxa* ZYPP18. (**A**) Colony morphology of ZYPP18 on LA cultured for 3 day; (**B**) Gram stain; (**C**) Spore character; (**D**–**F**) Phylogenetic tree based on the 16S rRNA gene, *gyrA* gene, and *rpoB* gene. (**D**) 16S rRNA gene; (**F**) *gyrA* gene; (**E**) *rpoB* gene.

**Figure 2 plants-12-02504-f002:**
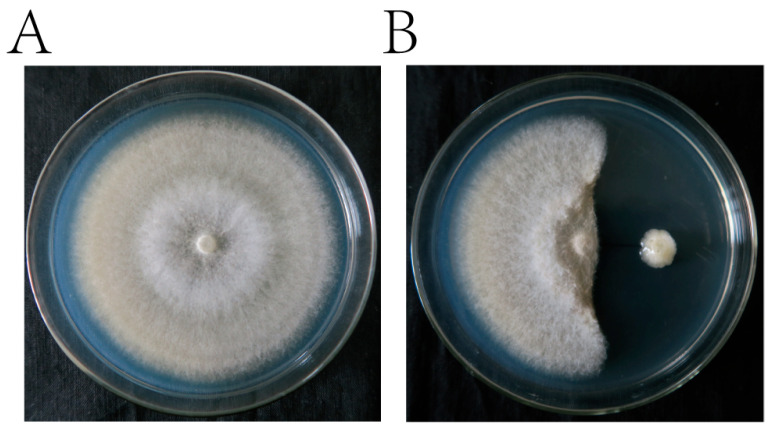
Antagonistic activity of *P. polymyxa* ZYPP18 against *R. cerealis.* (**A**) *R. cerealis* cultured on PDA for 7 d; (**B**) *R. cerealis* and ZYPP18 were co–cultured on PDA for 7 d.

**Figure 3 plants-12-02504-f003:**
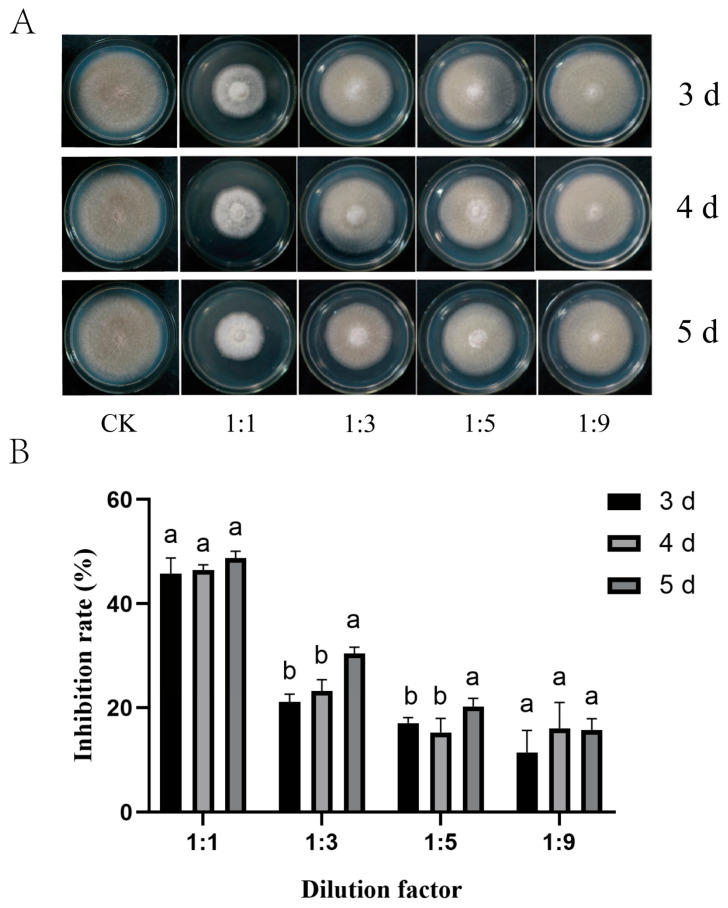
The inhibition effect of the fermentation filtrate of *P. polymyxa* ZYPP18 on the growth of *R. cerealis.* (**A**) the growth of *R. cerealis* on the PDA plates (CK) and the plates of PDA mixed with different ratios of the fermentation filtrate of ZYPP18 cultured in LB for 3 d, 4 d, and 5 d; (**B**) inhibition rate. Bar diagrams represent the mean of three replicates. Error bars indicate the standard deviation (SD) of the three replicates. Different lowercase letters mean the fermentation filtrate of inhibition rate cultured for 3 d, 4 d, and 5 d there were significant differences (*p* < 0.05).

**Figure 4 plants-12-02504-f004:**
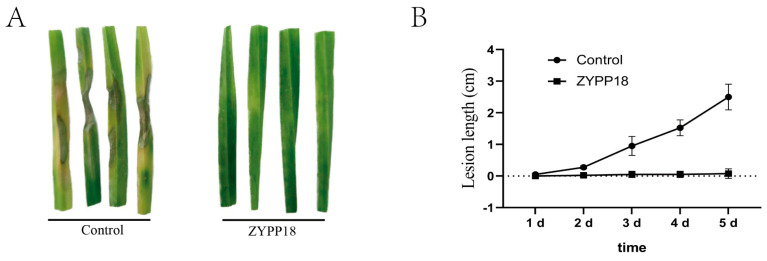
Inhibition effect of *P. polymyxa* ZYPP18 on wheat sheath blight expansion on detached leaves. (**A**) the disease expansion 5 d after the inoculation of *R. cerealis* on the detached leaves sprayed with sterile water (control) or a bacterial suspension of *P. polymyxa* ZYPP18 (ZYPP18); (**B**) the lesion length of wheat sheath blight on the detached wheat leaves. The line graph indicates the mean of the four replicates. Error bars indicate the standard deviation (SD) of the four replicates.

**Figure 5 plants-12-02504-f005:**
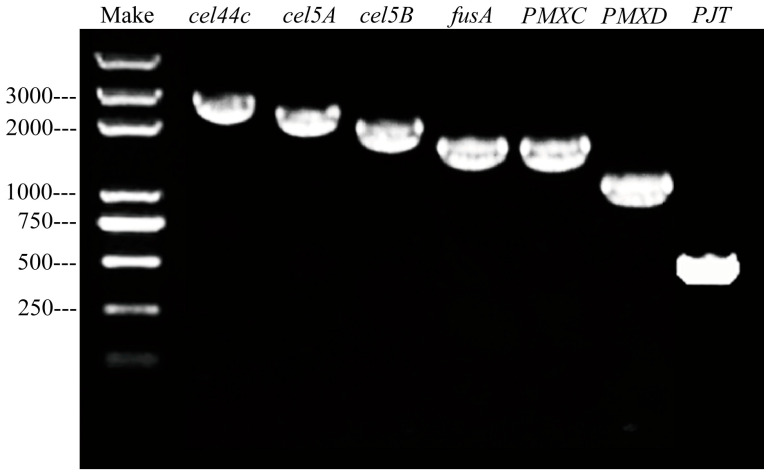
Detection of the genes related to antibacterial substance synthesis in *P. polymyxa* ZYPP18.

**Figure 6 plants-12-02504-f006:**
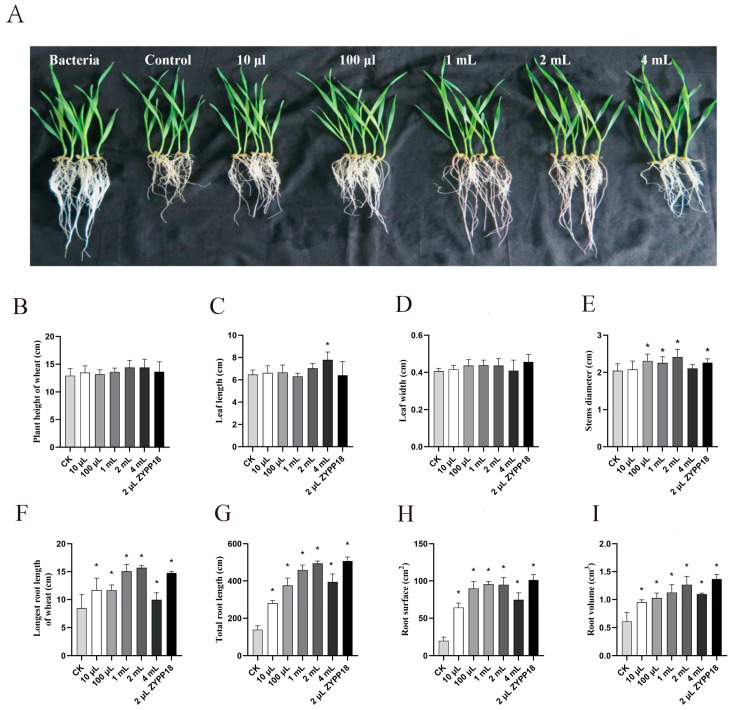
The growth–promoting effect of *B. polymyxa* ZYPP18 on wheat seedlings. (**A**) wheat seedlings grown in the Hoagland nutritional fluid with ZYPP18 or different contents of fermentation filtrate of ZYPP18; (**B**–**I**) the growth–promoting effect when treated with ZYPP18 or different contents of fermentation filtrate. (**B**) Longest root length of wheat, (**C**) Leaf length; (**D**) Leaf width; (**E**) Stems diameter; (**F**) Longest root length of wheat; (**G**) Total root length; (**H**) Root surface; (**I**) Root volume. Bar diagrams represent the mean of five replicates. Error bars indicate the standard deviation (SD) of the five replicates. Bar diagrams marked with “*” indicate significant differences between the treatment and CK (*p* < 0.05).

**Figure 7 plants-12-02504-f007:**
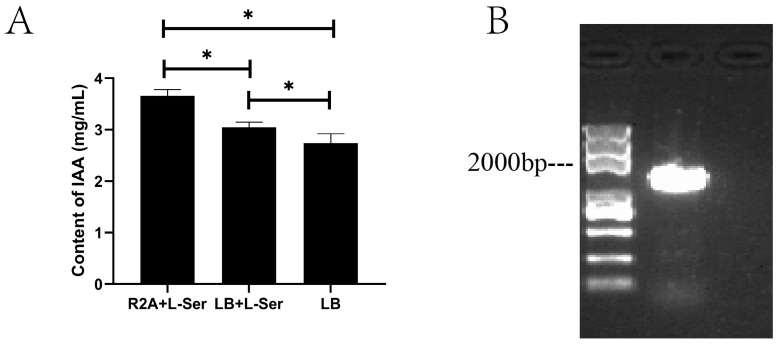
IAA detection in *B. polymyxa* ZYPP18. (**A**) Determination of the IAA content; (**B**) IAA–encoding gene detection. Bar diagrams represent the mean of three replicates. Error bars indicate the standard deviation (SD) of the three replicates. Bars diagrams marked with “*” indicate significant differences between different culture mediums (*p* < 0.05).

**Figure 8 plants-12-02504-f008:**
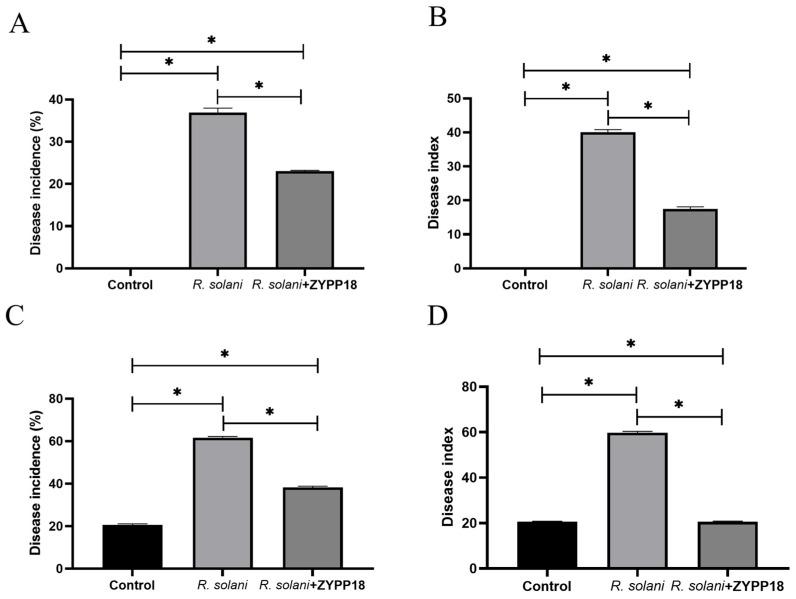
Disease incidence and disease index of wheat sheath blight in pot and field experiments. (**A**) Disease incidence in pot experiment; (**B**) Disease index in pot experiment; (**C**) Disease incidence in field experiment; (**D**) Disease index in field experiment. Bar diagrams represent the mean of three replicates. Error bars indicate the standard deviation (SD) of the three replicates. Bars diagrams marked with “*” indicate significant differences between treatments and CK or between treatments (*p* < 0.05).

**Table 1 plants-12-02504-t001:** Phosphorus solubilization activity of *B. polymyxa* ZYPP18.

Phosphorus	Colony Diameter of ZYPP18 (d) (mm)	Transparent Circle (D) (mm)	Transparent Circle/ZYPP18 Diameter (D/d)
Organic phosphorus	8.3 ± 0.6	12.0 ± 0.0	1.4 ± 0.1
Inorganic phosphorus	13.3 ± 1.5	20.7 ± 0.6	1.6 ± 0.1

**Table 2 plants-12-02504-t002:** Acid–base neutral phosphatase activity of *B. polymyxa* ZYPP18.

Phosphatase	Phosphatase Activity of ZYPP18 Cultured for 3 d (mg/mL^−1^, 37 °C, 24 h)	Phosphatase Activity of ZYPP18 Cultured for 5 d (mg/mL^−1^, 37 °C, 24 h)
Acid phosphatase	3.014 ± 0.004	1.305 ± 0.002
Alkaline phosphatase	13.846 ± 0.007	15.972 ± 0.002
Neutral phosphatase	0.473 ± 0.001	0.461 ± 0.003

Explanatory note: the data denote the mean ± SD (*n* = 3).

**Table 3 plants-12-02504-t003:** Control effect of *B. polymyxa* ZYPP18 on wheat sheath blight.

Experiments	Control Effect (%)
Pot experiments	56.30 ± 0.55
Field experiments	65.57 ± 0.22

Explanatory note: the data denote the mean ± SD (*n* = 3).

**Table 4 plants-12-02504-t004:** Primers used for the detection of the genes related to antimicrobial substance synthesis.

Antibiotic	Primer	Sequence 5′–3′
Polymyxin C	*PMXC*–F	TACACGGTCTTTGTGGTCATC
	*PMXC*–R	GCGTACAGTCCCTTCATCTTC
Polymyxin D	*PMXD*–F	CTCGCAGGTTTACTTCGTTT
	*PMXD*–R	CCAATGCTGGGATTCGTTAT
Fusaricidins	*fusA*–F	CAAGGATTCGACCGTAGGTG
	f*usA*–R	GTAGGGATTATGGCTGACCG
Hydrolysis of proteases	*cel44C*–F	TTTGGTTACCGCATGGGGTG
	*cel44C*–R	TTTCGGACGGAGAGGAGAGTGT
β–Glucanase	*PJT*–F	TACTAATTGCTCGTATATTTTACCCA
	*PJT*–R	TTGCGAATGTGTTCTGGGAACC
Cellulase A	*cel5A*–F	CTGCTCAACCTGGTCAACG
	*cel5A*–R	GCTCAAGGGCATTAGTTCTC
Cellulase B	*cel5B*–F	CTTGCTGTTGGCATTGAGC
	*cel5B*–R	CCTTTGCGAATCCATCTTTC

## Data Availability

Not applicable.

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
