# Peer review of "Characterization of Antagonistic Bacteria Paenibacillus polymyxa ZYPP18 and the Effects on Plant Growth"

_plants, 2023, doi:10.3390/plants12132504_

Round 1
Reviewer 1 Report
Journal: Plants
Manuscript ID: plants-2450800
Manuscript type: Article
Title: Characterization of Antagonistic Bacteria Paenibacillus polymyxa ZYPP18
and the Effects on Plant Growth
General information:
Generally, it can be an interesting manuscript about Paenibacillus ZYPP18 and its growth promoting abilities. However, I am afraid that I am not able to make a thorough and sufficient review on this version of the manuscript. The manuscript is missing all literature, which makes it impossible to give any final recommendation.
Additionally, the language of the manuscript could be improved in all sections, I was not able to understand how the pot experiment was performed. I think that short description of bacterial phosphatase activity methods is needed. Please also check my following comments, which I hope will be useful to improve your manuscript clarity.
Please rearrange the order of sections according to Plants requirements.
Please indicate how the specificity of used primers were confirmed the bands for polymyxins are quite strong, but the bands for remaining genes are much weaker in comparison. Please add the positive and negative controls in the supplementary data.
Generally, 16S sequencing is now considered as insufficient method for assignment of isolates to species level.
In text comments:
Line 11-12: Too general statement, please list the proven modes of action.
Line 13: Rhizoctonia solanis
Line 14: increasingly serious – please use more precise term.
Line 16: Too general: consider using: artificial growth media.
Line 17: detached leaf assay
Line 21: Please be more precise, how was the control efficiency been measured.
Line 28: PGPR are plural, and again too general statement, please give reference to the activities including but root growth promotion, Induced systematic Resistance, and suppression of pathogen growth by multiple modes of action.
Line 33: Please give the names of the disease caused by this pathogen.
Line 34: “somewhat” this term rarely finds use in scientific publication, please use more precise terminology.
Line 42: IAA the first use of this abbreviation please give a full name.
Line 59: I hope, it is the Lysogenic Agar that was inoculated not the serial dilutions.
Line 60: I would not recommend growing the isolates which are to be used for biological plant protection in that temperature. Consider using the same temperature as in following tests in the future experiments.
Line 75: I hope the isolates were plated on the same petri dish 2cm from the R. cerealis mycelium.
Line 79: Please try to be more precise, as far as I can understand this method the negative control has been cultured on the richer medium, which can be a reason for better growth.
Line 91: sterilization with only alcohol is usually not efficient, consider using hypochlorite or other more potent sterilizing agents. What was the medium for bacteria suspension, consider using the same sterile medium as negative control instead of water.
Line 148: The explanation for the pot experiment is not clear.
Line 279 I understand the there is no statistically significant difference in the disease incidence between plant treated with pathogen alone and pathogen + P. polymyxa
Line 296 Paenibacillus
Please consider improving the language in all sections, there are multiple mental shortcuts in your manuscript and some of them are hard to follow. Please try to consult a person not involved in your study to improve the methods description.
Reviewer 2 Report
- How many replicate you in all experiments
- Tables and Fig did not clear very will e.g. what the bar mean and what the letter on the columan
in table what is the +- is Sd or SE
- When you isolated the pathogen only you found one isolate?
- Discussion part should be improve with more details
- Make sure that all scientific names in the References list are italics.
- Please add the DOI for ALL the References
The main problem of this MS is authors did not write anything about the experiment design and statistical analysis and if they repeated the experiment or not therefore I cannot accept this paper to be publish
Moderate editing of English language required
Round 2
Reviewer 1 Report
Generally, the manuscript has improved and most of my comments are properly answered. I am glad that you could confirm the species assignment with additional two genes this should be sufficient. For the IAA production experiment, you explain that you use media supernatants therefore non inoculated medium should be used as negative control. This artificial media gives low positive results after prolonged incubation with the Salkowski reagent. Please just confirm it by incubating non inoculated medium the same amount of time with the reagent. This is extremely important to include appropriate control for your experimental design.
Figure 3: please include the name and parameters of used statistics.
Statistical analysis: I am concerned if One-way ANOVA could be used for your data, please confirm that the test requirements are meet and use appropriate test for your data. Beside p-value please give the test results (F for ANOVA) but also appropriate post hoc test results.
I consider this manuscript acceptable for publication after minor revisions.
Reviewer 2 Report
I have not any comments
Minor editing of English language required
